# Effect of Lemon (*Citrus limon*, L.) Peel Powder on Oocyst Shedding, Intestinal Health, and Performance of Broilers Exposed to *E. tenella* Challenge

**DOI:** 10.3390/ani13223533

**Published:** 2023-11-15

**Authors:** Abdul Hafeez, Israr Ahmad, Shabana Naz, Rasha Alonaizan, Rasha K. Al-akeel, Rifat Ullah Khan, Vincenzo Tufarelli

**Affiliations:** 1Department of Poultry Science, Faculty of Animal Husbandry and Veterinary Sciences, The University of Agriculture, Peshawar 25000, Pakistan; 2Department of Zoology, Government College University, Faisalabad 38000, Pakistan; 3Department of Zoology, College of Science, King Saud University, Riyadh 11451, Saudi Arabia; 4College of Veterinary Sciences, Faculty of Animal Husbandry and Veterinary Sciences, The University of Agriculture, Peshawar 25000, Pakistan; 5Department of Precision and Regenerative Medicine and Jonian Area, Section of Veterinary Science and Animal Production, University of Bari Aldo Moro, 70010 Valenzano, Italy

**Keywords:** lemon peel powder, broilers, coccidiosis lesion score, cecum histology

## Abstract

**Simple Summary:**

This study delved into the promising application of repurposed lemon peel powder as a supplement for augmenting the growth rate and bolstering the intestinal condition in broilers. Furthermore, it aimed to evaluate its potential as a natural alternative against experimental coccidiosis. The results revealed that incorporating 3 and 6 g/kg of lemon peel powder into the diet effectively countered the negative effect of coccidial oocysts on the birds’ growth rates. This intervention resulted in notable reductions in cecal lesion scores and a reduction in oocyst shedding. Furthermore, it played a pivotal role in revitalizing the cecal villi of the broilers. These findings underscore the considerable promise of lemon peel powder as a viable and natural means to alleviate the adverse effects of coccidiosis and to enhance the overall well-being and performance of broiler chickens.

**Abstract:**

To date, no study has reported the anticoccidial effect of lemon peel powder in broilers. Coccidiosis, caused by *Eimeria* species, is the prevalent enteric parasitic disease in poultry. Although certain chemical drugs have been used for their control, concerns regarding drug residues and the development of resistance in chickens have arisen among consumers. In this study, a total of 300 Ross 308 broiler chicks were randomly allocated into five groups (five equal replicates of 12 animals). The first group served as the control and did not receive any specific treatment (NC). The second group, referred to as the positive control (PC) group, was deliberately exposed to *Eimeria tenella*. The third group was challenged with *E. tenella* and also received treatment with amprolium (1 g/kg) and was designated as AT. The fourth and fifth groups were challenged with *E. tenella* and simultaneously supplemented with lemon peel powder at a dosage of 3 g/kg (LPP3) and 6 g/kg (LPP6). Sporulated *E. tenella* oocysts (5 × 10^4^/mL bird) on day 22 of the experiment were administered to the infected broiler chickens. The results indicated that in comparison with the NC, all *Eimeria*-treated birds exhibited significantly (*p* < 0.05) lower growth performance. However, a notable improvement was observed when infected birds also received a supplement of LPP3 and LPP6 in their feed. Both LPP3 and LPP6 supplementation significantly (*p* < 0.05) reduced mortality, lesion scores, and oocyst per gram (OPG) of feces compared with the PC group. Additionally, the histological features of the cecum revealed that villus height, villus width, and crypt depth were partially restored under supplementation with LPP3 and LPP6 in the infected birds. Overall, the results demonstrate that *Eimeria*-infected birds supplemented with LPP3 and LPP6 exhibited improved growth performance, reduced OPG, lowered intestinal coccidiosis lesion scores, and enhanced intestinal histological features.

## 1. Introduction

Coccidiosis is an economically important parasitic disease caused by protozoa, characterized by the presence of lesions in the intestine [1]. It is one of the most expensive parasitic diseases affecting poultry production as a result of various species of *Eimeria*. Typically, *Eimeria* species multiply and induce severe alternations in the epithelial tissue of different parts of the intestine through multiplication [2]. The economic impact of coccidiosis is substantial due to its widespread global occurrence in countries with significant poultry production [3]. Economic losses associated with this disease encompass weight reduction, diminished feed efficiency, increased costs of anti-coccidial therapy, and high mortality and morbidity [2,4]. Moreover, *Eimeria* infections can disrupt digestion and absorption of important nutrients, resulting in impaired performance and immunosuppression [5].

In the present scenario, the primary approach against coccidiosis in poultry farms involves the application of chemical drugs, vaccination, or natural compounds [2,6]. Although using preventive chemical agents has proven highly effective in controlling parasitic infections, concerns have arisen due to the development of drug resistance in poultry products [7,8]. As a result, researchers have been actively seeking natural alternative health-conscious solutions [9,10]. However, in developed countries, the use of anti-coccidial drugs for preventing coccidiosis has been severely restricted for more than one decade and is anticipated to be completely withdrawn in the upcoming years [11].

A number of research studies have investigated the positive impacts of plant-based compounds on broiler performance, gastrointestinal function, and mortality. In coccidiosis control programs, phytogenic compounds present themselves as a viable alternative to chemical counterparts. These plant-based anti-coccidial compounds not only exhibit inhibitory effects against *Eimeria* species but also tend to have minimal adverse effects on the health of birds [12]. Recent reports have highlighted the anti-coccidial properties of phytogenic compounds, including plant extracts, spices, and essential oils [6]. While the use of phytogenic products represents an important option for preventing coccidiosis in antibiotic-free poultry production systems [3], their wider adoption across poultry farms may be encouraged if these compounds originate from agricultural by-products, providing a cost-effective and more potent alternative to chemical options [6]. 

The positive impact of citrus peel supplementation in poultry diets has been documented in various aspects including enhanced performance and nutrient digestibility and improved antioxidant enzyme profiles, as well as favorable alterations in blood metabolites [13,14,15,16,17]. The pulp of lemons (*Citrus limon*) is a common by-product in the food and juice extraction industry, and lemons are the most widely consumed citrus fruit globally [18,19]. Citrus fruit pulp accounts for about one-fourth of the total fruit mass and is obtained after extracting the juice and mechanically removing the remaining pulp [20]. During the extraction process of citrus juice, a significant amount of waste or by-products is generated [21]. Ramadan et al. [22] found that lemon peel extracts exhibit antimicrobial activity, suggesting their potential use as sanitizers to mitigate microbial contamination from foodborne bacteria. While a substantial quantity of citrus pulp is utilized as feed for animals (particularly ruminants), the majority of the processing residue is discarded, resulting in environmental pollution [23]. As a result, citrus-processing industries have been actively seeking alternative applications for these by-products [24]. A number of studies have reported beneficial effects of citrus such as improved growth performance, carcass quality, antioxidant activity, intestinal histomorphology, and other serum metabolites in broilers [24,25,26]. However, to the best of our knowledge, no study has reported the anticoccidial effect of lemon peel powder in broilers. Therefore, this study was conducted to evaluate different levels of lemon peel powder dietary supplementation on the growth performance, intestinal lesion score, mortality rate, oocysts shedding, and intestinal histomorphology in broilers experimentally exposed to *Eimeria tenella* challenge. 

## 2. Materials and Methods 

### 2.1. Lemon Peel Powder (LPP) Preparation

About 3 kg of lemons (*Citrus limon*, L.) were purchased from the local market. Following the extraction of the edible portion of the lemon fruit, about 1 kg of lemon peels was air dried at room temperature in a shaded area and subsequently ground into a powdered form of about 950 g after removing the residues and big particles.

### 2.2. Preparation of Eimeria Oocysts

*Eimeria* spp. oocysts were obtained from fecal samples of naturally infected broilers. These collected oocysts were combined with a 2.5% potassium dichromate solution and then incubated at 25 °C with aeration. Daily assessments were conducted to monitor the sporulation progress [6]. Once sporulation was complete, the oocysts were subjected to multiple centrifugation washes until the supernatant became clear. The clear supernatant was then extracted and stored at 4 °C for subsequent use. The oocysts were cultured, isolated, and sporulated using the method outlined by Raether et al. [27]. This involved a series of density gradient centrifugation steps in a glucose solution to separate oocysts of varying densities. The purified oocysts were thoroughly washed to eliminate any residual impurities. Following washing, they were re-suspended in an appropriate solution. A minute sample of these purified oocysts underwent microscopic examination. *Eimeria tenella* oocysts exhibit distinctive features enabling their precise identification. For the inoculation process, the oocysts suspended in potassium dichromate were rinsed twice with distilled water. Subsequently, they were adjusted to a concentration of 10,000 oocysts per mL of distilled water using the enhanced McMaster method [28]. The infection was introduced on the 22nd day of the experiment and lasted until day 35 of the experiment. 

To prevent cross-contamination, separate plastic boots were designated for the challenged and non-challenged pens. The challenged group had exclusive boot use, while the non-challenged had their set. Additionally, calcium oxide trays at non-challenged pen entrances acted as footbaths, adding an extra barrier against contamination. 

### 2.3. Birds Husbandry and Experimental Design

In this study, a total of 300 Ross 308 broiler chicks were randomly assigned into five groups (each with five equal replicates). All the chickens were raised under identical nutritional and management conditions until they reached 21 days of age. They were subjected to a continuous lighting schedule and had free access to both feed (Table 1) and water (ad libitum). The 1st group, designated as the control, did not receive any specific treatment (NC). The 2nd group, referred to as the positive control (PC) group, was deliberately exposed to *Eimeria* tenella. The 3rd group was infected with *E. tenella* and also received treatment with amprolium (100–150 g/100 L) from Huvepharma Inc., Peachtree City, GA, USA, and was designated as AT. To investigate the potential effects of specific supplements, the 4th group was challenged with *E. tenella* and simultaneously supplemented with lemon peel powder at a dosage of 3 g/kg (LPP3). The 5th group was similarly challenged with *E. tenella* and received a higher supplementation of lemon peel powder at a dosage of 6 g/kg (LPP6) of feed. 

### 2.4. Coccidial Infection 

All treatment groups were orally administered with sporulated *E. tenella* oocysts (5 × 10^4^/mL/bird) on day 22 of the experiment. Stringent measures were implemented to prevent any contact between infected and uninfected birds, even though they were housed in close proximity. Mortality was recorded for each replicate after the *Eimeria* challenge. The “NC” group received an equivalent volume of normal saline through gavage.

### 2.5. Performance Traits

Throughout the entire growth period, we assessed key growth indices including feed intake (FI), weight gain (WG), and the feed conversion ratio (FCR) on a weekly basis.

### 2.6. Lesion Score

From the removal of the cecum onward, a thorough evaluation of its lesions was conducted. A detailed grading scale ranging from 0 (no lesions) to 4 (very severe lesions) was used, aligning with the criteria established by Khorrami et al. [2]. This meticulous approach ensured a comprehensive assessment of cecal health.

### 2.7. Oocysts Counting

The combined fecal samples from each cage were assessed at three specific time points (7, 10, and 14 days) after infection (dpi), following the methodology outlined by Khan et al. [1]. A clean plastic sheet was positioned beneath the wired cage floor for a duration of 24 h. All fecal droppings that accumulated on the sheet were gathered and weighed. Following this, the feces were thoroughly combined, and a 4 g sample was extracted for subsequent processing. To quantify oocyst production, a modified McMaster counting chamber technique, as outlined by Long and Rowell (1958), was used. In short, 20 mL of a sodium chloride solution was introduced to the fecal sample. Following thorough mixing, 2 mL of the resulting suspension was transferred into a centrifuge tube containing 8 mL of a saturated sodium chloride solution. This suspension was then used to fill two McMaster counting chambers. Oocysts per gram (OPG) were quantified using a McMaster chamber and counted using a compound microscope (Nikon, Tokyo, Japan) under low magnification.

### 2.8. Pathological Alterations

Two cecal samples per replicate were aseptically processed for histological analysis [1]. This involved excising cecal tissue sections measuring 1 cm^2^, which were then placed in a 10% buffered formalin solution. Subsequently, the tissue samples underwent dehydration using a freshly prepared alcohol solution. Using a microtome (Accu-Cut SRM 200 Sakura, Finetek, The Netherlands), tissue blocks were sliced and subjected to H and E staining and then examined under a microscope (Nikon, Japan) at 40× magnification for detailed observations. 

### 2.9. Statistical Analysis

The data underwent an analysis of variance (ANOVA), and for assessing the significance of means, Tukey’s test was applied, with a predetermined significance level of *p* < 0.05. STATISTIC-2010 (version 2.1) statistical software was the tool used for data analysis. The lesion score was determined using the nonparametric Kruskal–Wallis H test, while mortality was expressed as a percentage.

## 3. Results

Table 2 shows that supplementation with lemon peel powder had a significant effect on FI in broilers. At week 1, FI was significantly (*p* < 0.05) higher in the NC group and PC group than the LPP6 group, with no significant (*p* > 0.05) differences between the AT group and the LPP3 group. At week 2, FI was significantly (*p* < 0.05) higher in the NC group, PC group, and the AT group followed by the LPP3 group and the LPP6 group. FI at week 3 was significantly (*p* < 0.05) higher in the NC group, which had no significant difference with the PC group and the AT group, followed by the LPP3 group and the LPP6 group. In the starter phase, FI was significantly (*p* < 0.05) higher in the NC group, the PC group, and the AT group followed by the LPP3 group and the LPP6 group. At week 4, the highest FI was observed in the NC group in comparison with the AT group followed by the LPP6 group, the LPP3 group, and the PC group. At week 5, FI in the NC group was significantly (*p* < 0.05) higher than the AT group followed by the LPP6 group, the LPP3 group, and the PC group. FI in the finisher phase was significantly (*p* < 0.05) higher in the NC group than in the AT group followed by the LPP6 group, the LPP3 group, and the PC group. For the overall period, FI was significantly (*p* < 0.05) higher in the NC group in comparison with the AT group, followed by the LPP6 group, the LPP3 group, and the PC group. There was no significant (*p* > 0.05) difference between the lemon peel-supplemented groups in overall FI.

Table 3 illustrates the results of weight gain, which indicates that there was no significant difference in week 1, week 2, and the starter phase. At week 3, weight gain in the LPP6 group had no significant (*p* < 0.05) difference compared to the LPP3 group, the NC group, and the PC group but was significantly (*p* > 0.05) higher than the AT group. At week 4, weight gain in the NC group was significantly (*p* > 0.05) higher than the AT group followed by the LPP6 group, the LP3 group, and the PC group. At week 5, WG was significantly (*p* > 0.05) higher in the NC group as compared with the AT group, followed by the LPP6 group, the LPP3 group, and the PC group. In the finisher phase, weight gain in the NC group was significantly (*p* > 0.05) higher than the AT group followed by the LPP6 group, the LPP3 group, and the PC group. For the overall period, WG was significantly (*p* > 0.05) higher in the NC group than the AT group followed by the LPP6 group, the LPP3 group, and the PC group. There was no significant (*p* < 0.05) difference between the LPP3 and LPP6 groups in overall weight gain.

Table 4 illustrates the effect of adding lemon peel powder to the broiler diets on the FCR. At week 1, there was no significant (*p* > 0.05) difference in the FCR among the groups. At week 2, the FCR was significantly (*p* < 0.05) lower in the LPP6 group and LPP3 group compared with the AT, NC, and PC groups. At week 3, the FCR in the LPP6 group was similar to the LPP3 group followed by the NC group, PC group, and the AT group. In the starter phase, the LPP6 group and LPP3 group had a significantly (*p* < 0.05) lower FCR than the NC group and the PC group followed by the AT group. At week 4, the FCR of the NC group and the AT group was significantly (*p* < 0.05) lower than the LPP6 group and the LPP3 group followed by the PC group. At week 5, the FCR in the NC group was significantly (*p* < 0.05) lower in comparison with the AT group followed by the LPP6 group and the LPP3 group. Moreover, the FCR of the PC group was the highest (*p* < 0.05) in week 5. In the finisher phase, the FCR in the NC group was significantly (*p* < 0.05) lower than the AT group followed by the LPP6 group, the LPP3 group, and the PC group. For the overall period, the FCR of the NC group was significantly (*p* < 0.05) lower compared with the AT group and the LPP6 group followed by the LPP3 group and the PC group.

The effect of supplementation with lemon peel powder on the lesion score and mortality in the broiler diets challenged with coccidiosis is shown in Table 5. The result shows that the NC group, which was not infected, displayed a normal cecal epithelium. Cecal lesions were significantly (*p* < 0.05) higher in the PC group as compared with the LPP3 group, the LPP6 group, and the AT group. Cecal lesions were absent from the NC group, while the lowest lesions were observed in the AT-treated group. The mortality rate was higher in the PC group compared with the LPP3 and LPP6 groups. The lowest mortality was recorded in the NC and AT groups. 

Table 6 presents the effect of supplementation with lemon peel powder on oocyst OPG of feces in broilers challenged with coccidiosis. The results showed that OPG were significantly (*p* < 0.05) higher in the PC group compared with the NC and AT groups on 7, 10, and 14 DPI. No significant change was observed in OPG on 7 DPI between the PC, LPP3, and LPP6 groups. However, on day 10 and day 14 post-infection, OPG decreased significantly (*p* < 0.05) in the LPP3 and LPP6 groups compared with the PC group.

Table 7 shows the effect of supplementation of lemon peel powder on villus height, crypt depth, the villus height to crypt depth ratio, and the width of the caecum of broilers challenged with coccidiosis. Villus height and width and VH:CH decreased significantly (*p* < 0.05) in the PC group compared with the NC and AT groups. Interestingly, villus height and width and the ratio of VH and CD were significantly (*p* < 0.05) higher in the LPP3 and LPP6 groups compared with the PC group. Similarly, crypt depth decreased significantly (*p* < 0.05) in the LPP3 and LPP6 groups compared with the PC group. An examination of the cecal epithelium (Figure 1) revealed that in the PC group, prevalent signs observed in challenged and untreated chickens included the presence of blood, hemorrhages, and sloughing of the epithelium. Various types of inflammatory cells and *Eimeria* oocysts were also detected in the intestinal epithelium of these chickens. Conversely, both the infected groups supplemented with lemon peel powder (at 3 g/kg and 6 g/kg) exhibited fewer lesions and mild cecal sloughing. The group treated with amprolium after infection showed only a few lesions with minor damage to the intestinal epithelium. 

## 4. Discussion

The processing of lemons results in a significant amount of peel waste, which contains numerous high-value substances with substantial potential for industrial applications. Lemon peels are particularly abundant in valuable nutritional components, notably vitamin C, pectin, fibers, and various phytochemical compounds. These include phenolic constituents and essential oils, which contribute to the impressive nutraceutical potential of lemon peels [21]. Studies are scarce on the effect of lemon peel powder as a feed additive in the broiler diet. This is the first study to investigate the effect of LPP in broilers challenged with *E. tenella*. In this study, the impact of LPP on growth indicators, *Eimeria* oocyst excretion, and intestinal health in broiler chickens subjected to experimental *Eimeria* infection was examined. The findings demonstrate that LPP mitigates intestinal lesions following challenges with *Eimeria* oocysts and also leads to a reduction in the excretion of *Eimeria* oocysts from the gastrointestinal tract and partially restored cecal histopathological microscopic structures compared with the challenged and non-treated group. In the present study, the impact of LPP6 was similar to LPP3. In the present study, broilers challenged with *Eimeria* exhibited enhanced growth performance when supplemented with LPP3 and LPP6 compared with the challenged and non-treated group. Under no coccidial challenge, several studies have investigated the impact of lemon peel in different forms (extract, juice) on growth indices in chickens. Akbarian et al. [24] reported no significant effect on FI, weight gain, and the FCR in broilers fed diets supplemented with lemon peel extract (LPE) at 200 and 400 mg/kg under high ambient temperature. According to Ishaq et al. [26], a lower FCR was observed in groups treated with *C. aurantium* extract may be attributed to the extract’s ability to reduce intestinal inflammation and injuries in broilers. Similarly, Salehifar et al. [25] also reported no significant effect of 0.2%, 0.6%, and 1.0% lemon pulp powder on the growth performance of broilers under high ambient temperatures. Dudko et al. [29] found a significant effect on weight gain in Polish sheep supplemented with a blend of essential oil from *Origanum vulgare* (lamiaceae) and Citrus spp. (citraceae) under coccidiosis challenge. Ishaq et al. [26] reported that ethanolic leaf extract of *Citrus aurantium* at the levels of 125, 250, and 500 mg/kg resulted in higher daily FI but no change in weight gain in broiler chickens. The notable reduction in body weight gain observed in the PC group in the current study is similar to the outcome of previous reports. This phenomenon may be attributed to the negative implications for digest stability, absorption capacity, and nutritional assimilation. 

In the present study, both supplements demonstrated equal effectiveness in terms of lesion scores, oocysts per gram (OPG), and histological dimensions. Furthermore, the assessment of LPP3 and LPP6 doses revealed that while they exhibit some anti-coccidial effects at these concentrations, these effects are considerably less pronounced compared with AT in terms of mitigating typical coccidiosis symptoms or enhancing growth performance. Numerous studies have investigated the antimicrobial properties of lemon peels, demonstrating their broad-spectrum effectiveness [30]. Ishaq et al. [26] reported a reduction in oocysts shedding and improvement in the lesion score in broilers supplemented with 125, 250, and 750 mg/kg *Citrus aurantium* L. ethanolic leaf extract against experimental *E. tenella* infection. Ishaq et al. [26] suggested that the reduction in oocyst shedding observed with lemon peel extract may be attributed to its saponin content. This study indicates that saponins bind with sterol molecules on the parasite’s cell layer, leading to their destruction and subsequently reducing excretion in chicken feces. This study demonstrated a noteworthy reduction in cecal and lesion scores among the treated groups. This damage may be caused by the replication of developmental *Eimeria* parasites, potentially leading to inflammatory reactions and secondary bacterial infections, which could be associated with the observed lesions in the bird’s caeca [31]. Ishaq et al. [26] suggested that the decrease in oocyst shedding is attributed to the presence of antioxidant and free radical scavenging properties in essential oils from *C. aurantium* leaves likely contributed to an observed improvement in cecal lesions. Antioxidant compounds are recognized for their ability to diminish the number of harmful free radical molecules in the body, providing protection against their detrimental effects. In this study, lemon peel powder effectively reduced the mortality rate of broiler chickens compared with the challenged and non-treated group. The findings indicate that the lower oocyst excretion rate, increased survivability, reduced bloody diarrhea, and improved cecal condition in the LPP-treated groups, compared with the infected-untreated group, may be attributed to immune-enhancing compounds like tannins, alkaloids, and flavonoids present in lemon peels [21,22]. Phenolic extracts from lemon peels were found to effectively inhibit the growth of Staphylococcus aureus and Bacillus cereus. Recent research on citrus fruits revealed that lemon peel extract exhibited stronger antibacterial effects compared with pomelo peels but was slightly less potent than lime peels [32]. Ramadan et al. [22] reported that methanol and ethanol extracts from lemon peels are effective against several foodborne pathogens (*Escherichia coli*, *Salmonella Typhimurium*, *Listeria monocytogenes*) in in vitro tests using disc diffusion and minimum inhibition concentration assays, as well as in situ experiments on chicken skin. Treatment with 5 mg/mL of the extract significantly reduced *L. monocytogenes* and *P. fluorescens* levels. 

The presented study demonstrated that the deterioration in villus dimensions caused by *E. tenella* challenge was partially reversed by including LPP3 and LPP6 supplements in the broilers’ diets. The application of LPP treatment led to an improvement in the histological features of broilers infected with *Eimeria* challenge. This could be attributed to the antiprotozoal properties of lemon peel, as well as its potent antioxidant and anti-inflammatory activities. These attributes serve to safeguard host tissue from damage caused by *Eimeria* oocysts. Akbrarian et al. [24] reported an improvement in intestinal histological features in broilers supplemented with 200 and 400 mg/kg citrus peel extracts under high ambient temperatures. Similar improvements in histological structures were noted in broilers fed with 0.2%, 0.6%, and 1% lemon peel powder (LPP) in high ambient temperatures [25]. The effectiveness of lemon peels can be mainly attributed to their key chemical constituents, particularly vitamin C, fibers, pectin, and beneficial phytochemical compounds. These are due to the desirable properties of essential oils and phenolic constituents [21]. As natural products, lemon peels show great potential as sources for novel anticoccidial agents that target *Eimeria* while also providing protective and healing properties for infected host tissues. 

## 5. Conclusions

From the results of the present study, it was concluded that lemon peel powder at 3 and 6 g/kg improved growth performance, reduced excretion of *Eimeria* oocysts in the feces, and partially reversed histopathological deformities in broilers challenged with *E. tenella* infection compared with the challenged and non-treated group.

## Figures and Tables

**Figure 1 animals-13-03533-f001:**
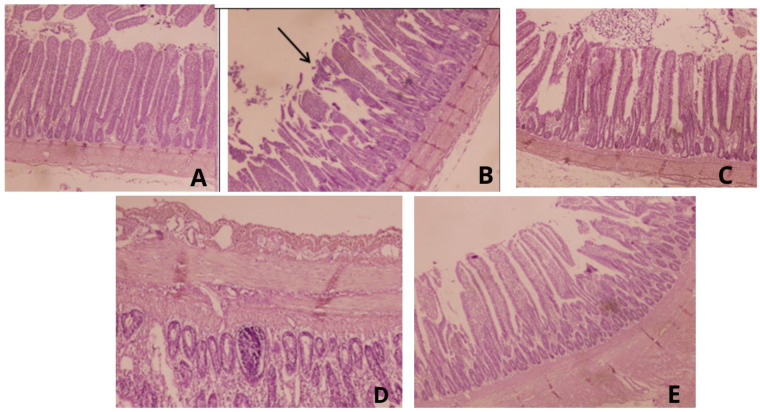
(**A**) Photomicrograph of cecal villus in broilers; NC shows normal villi. (**B**) Photomicrograph of cecal villus in broilers; (PC) displaying erosion and shedding of crypt epithelia, accompanied by the infiltration of responsive cells in the lamina propria. (**C**) Photomicrograph of cecal villus in broilers challenged with *E. tenella* + amprolium-treated, showing almost normal villi. (**D**,**E**) Intestines from broilers in LPP-treated groups (3 and 6 g/kg respectively) showing a mild sloughing of the epithelium and less infiltration in *Eimeria*-infected epithelium (HE ×200).

**Table 1 animals-13-03533-t001:** Feed ingredients and chemical analysis.

Ingredients	Starter Phase	Finisher Phase
Yellow corn	53.22	60.75
Soybean meal (48% CP)	37.92	25.00
Corn gluten meal (34% CP)	2.00	7.10
Corn oil	2.20	2.80
Dicalcium phosphate	2.30	2.05
Limestone	0.83	0.68
NaCl	0.45	0.50
Vitamin and minerals premix	0.50	0.50
DL-methionine	0.20	0.10
L-lysine HCl	0.22	0.37
L-threonine	0.11	0.10
Choline chloride	0.05	0.05
Chemical composition		
ME, kcal/kg	3000	3150
Crude protein, %	22.5	21.30
Methionine, %	0.55	0.44
Lysine, %	1.42	1.23
Sulfur amino acids, %	0.96	0.80
Threonine, %	0.95	0.85
Calcium, %	1.05	0.90
Available phosphorus, %	0.50	0.45

**Table 2 animals-13-03533-t002:** Effect of supplementation with lemon peel powder on feed intake (g) in broiler diets challenged with coccidiosis.

	NC	PC	AT	LPP3	LPP6	SEM	*p*-Value
Week 1	124.3 ^a^	126.7 ^a^	122.3 ^ab^	121.7 ^ab^	119.3 ^b^	0.87	0.05
Week 2	371.0 ^a^	369.7 ^a^	370.0 ^a^	364.7 ^ab^	360.3 ^b^	1.28	0.01
Week 3	611.7 ^a^	609.7 ^ab^	609.3 ^ab^	605.3 ^bc^	602.7 ^c^	1.01	0.01
Starter phase	1107.0 ^a^	1106.0 ^a^	1101.7 ^a^	1091.7 ^b^	1082.3 ^b^	2.64	<0.01
Week 4	725.3 ^a^	589.3 ^e^	708.7 ^b^	677.3 ^d^	685.7 ^c^	12.6	<0.01
Week 5	917.7 ^a^	765.3 ^d^	895.3 ^b^	879.7 ^c^	884.3 ^c^	14.2	<0.01
Finisher phase	1643.0 ^a^	1354.7 ^e^	1604.0 ^b^	1557.0 ^d^	1570.0 ^c^	26.8	<0.01
Overall mean	2750.0 ^a^	2460.7 ^d^	2705.7 ^b^	2648.7 ^c^	2652.3 ^c^	26.4	<0.01

Means in the same row with dissimilar superscripts are significantly different at (*p* < 0.05). Groups: NC = (negative control), PC = (infected + untreated), AT = (infected + amprolium-treated), LPP3 = (infected + treated with 3 g/kg lemon peel powder), LPP6 = (infected + treated with 6 g/kg lemon peel powder).

**Table 3 animals-13-03533-t003:** Effect of supplementation with lemon peel powder on weight gain (g) in broiler diets challenged with coccidiosis (g).

Group	NC	PC	AT	LPP3	LPP6	SEM	*p*-Value
Week 1	112.3	111.7	112.3	109.3	107.7	0.71	0.13
Week 2	280.7	279.3	282.7	281.7	284.7	0.86	0.38
Week 3	388.3 ^ab^	386.3 ^ab^	384.7 ^b^	391.0 ^ab^	393.3 ^a^	1.06	0.03
Starter phase	781.3	777.3	779.7	782.0	785.7	1.28	0.36
Week 4	466.7 ^a^	295.3 ^d^	447.0 ^b^	416.3 ^c^	418.3 ^c^	16.0	<0.05
Week 5	508.0 ^a^	359.7 ^e^	472.7 ^b^	435.0 ^d^	447.3 ^c^	13.2	<0.05
Finisher phase	974.7 ^a^	655.0 ^e^	919.7 ^b^	851.3 ^d^	865.7 ^c^	29.0	<0.05
Overall mean	1756.0 ^a^	1432.3 ^d^	1699.3 ^b^	1633.3 ^c^	1651.3 ^c^	29.4	<0.05

Means in the same row with dissimilar superscripts are significantly different at (*p* < 0.05). Groups: NC = (negative control), PC = (infected + untreated), AT = (infected + amprolium-treated), LPP3 = (infected + treated with 3 g/kg lemon peel powder), LPP6 = (infected + treated with 6 g/kg lemon peel powder).

**Table 4 animals-13-03533-t004:** Effect of supplementation with lemon peel powder on the feed conversion ratio of broilers challenged with coccidiosis.

Group	NC	PC	AT	LPP3	LPP6	SEM	*p*-Value
Week 1	1.11	1.13	1.09	1.11	1.11	0.01	0.171
Week 2	1.32 ^a^	1.33 ^a^	1.31 ^a^	1.29 ^ab^	1.27 ^b^	0.01	0.007
Week 3	1.58 ^ab^	1.58 ^ab^	1.59 ^a^	1.55 ^bc^	1.53 ^c^	0.01	0.004
Starter	1.42 ^ab^	1.42 ^ab^	1.42 ^a^	1.39 ^bc^	1.38 ^c^	0.01	0.001
Week 4	1.55 ^c^	1.99 ^a^	1.59 ^c^	1.63 ^b^	1.64 ^b^	0.04	<0.05
Week 5	1.81 ^e^	2.13 ^a^	1.89 ^d^	2.02 ^b^	1.98 ^c^	0.03	<0.05
Finisher	1.68 ^d^	2.07 ^a^	1.74 ^c^	1.83 ^b^	1.81 ^b^	0.04	<0.05
Overall	1.57 ^d^	1.71 ^a^	1.59 ^c^	1.62 ^b^	1.61 ^bc^	0.01	<0.05

Means in the same row with dissimilar superscripts are significantly different at (*p* < 0.05). Groups: NC = (negative control), PC = (infected + untreated), AT = (infected + amprolium-treated), LPP3 = (infected + treated with 3 g/kg lemon peel powder), LPP6 = (infected + treated with 6 g/kg lemon peel powder).

**Table 5 animals-13-03533-t005:** Effect of supplementation with lemon peel powder on lesion score and mortality for broiler diets challenged with coccidiosis.

Groups	Lesion Scoring	Mortality (%)
NC	0.00 ^c^	1.3 ^c^
PC	3.00 ^a^	18.70 ^a^
AT	0.67 ^c^	1.50 ^c^
LPP3	2.17 ^b^	11.64 ^b^
LPP6	2.13 ^b^	11.00 ^b^
*p*-value	0.00	0.012

Means in the same column with dissimilar superscripts are significantly different at (*p* < 0.05) Groups: NC = (negative control), PC = (infected + untreated), AT = (infected + amprolium-treated), LPP3 = (infected + treated with 3 g/kg lemon peel powder), LPP6 = (infected + treated with 6 g/kg lemon peel powder). Means in the same row with dissimilar superscripts are significantly different at (*p* < 0.05).

**Table 6 animals-13-03533-t006:** Effect of supplementation with lemon peel powder on oocyst count per gram of feces in broilers challenged with coccidiosis.

Groups	5 DPI	7 DPI	9 DPI
NC	00 ^c^ ± 00	00 ^c^ ± 0.00	00 ^b^ ± 0.00
PC	241.13 ^a^ ± 14.32	543.31 ^a^ ± 19.87	282.79 ^a^ ± 11.57
AT	99.42 ^b^ ± 3.56	188.170 ^b^ ± 18.98	104.83 ^b^ ± 11.73
LPP3	145.45 ^ab^ ± 8.57	222.52 ^b^ ± 5.78	126.52 ^b^ ± 2.49
LPP6	148.48 ^ab^ ± 9.18	219.46 ^b^ ± 18.91	129.46 ^b^ ± 5.89
*p*-value	0.0015	0.014	0.019

Means in the same column but with dissimilar superscripts are significantly different at *p* < 0.05. Groups: NC = (negative control), PC = (infected + untreated), AT = (infected + amprolium-treated), LPP3 = (infected + treated with 3 g/kg lemon peel powder), LPP6 = (infected + treated with 6 g/kg lemon peel powder). Means in the same row with dissimilar superscripts are significantly different at (*p* < 0.05). DPI: day post-infection.

**Table 7 animals-13-03533-t007:** Effect of supplementation with lemon peel powder on villus height, crept depth, the villus height to crypt depth ratio, and width of the caecum of broilers challenged with coccidiosis.

Groups	Villus Height (mm)	Crypt Depth (mm)	VH:CD	Villus Width (mm)
NC	0.49 ^a^ ± 0.12	0.11 ^c^ ± 0.01	6.29 ^a^ ± 0.39	0.25 ^a^ ± 0.01
PC	0.24 ^c^ ± 0.11	0.19 ^a^ ± 0.01	2.03 ^c^ ± 0.13	0.15 ^c^ ± 0.02
AT	0.42 ^a^ ± 0.11	0.12 ^c^ ± 0.01	5.11 ^a^ ± 0.24	0.21 ^ab^ ± 0.01
LPP3	0.29 ^ab^ ± 0.11	0.15 ^b^ ± 0.01	3.48 ^b^ ± 0.38	0.18 ^b^ ± 0.01
LPP6	0.28 ^b^ ± 0.11	0.15 ^b^ ± 0.01	3.49 ^b^ ± 0.31	0.18 ^b^ ± 0.01
*p*-value	0.0003	0.0234	0.0404	0.025

Means in the same column with dissimilar superscripts are significantly different at *p* < 0.05. Groups: NC = (negative control), PC = (infected + untreated), AT = (infected + amprolium-treated), LPP3 = (infected + treated with 3 g/kg lemon peel powder), LPP6 = (infected + treated with 6 g/kg lemon peel powder). Means in the same row with different superscripts are significantly different at (*p* < 0.05).

## Data Availability

Data are contained within the article.

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
