# Peer review of "Effect of Lemon (Citrus limon, L.) Peel Powder on Oocyst Shedding, Intestinal Health, and Performance of Broilers Exposed to E. tenella Challenge"

_animals, 2023, doi:10.3390/ani13223533_

Round 1

Reviewer 1 Report

Comments and Suggestions for Authors

The conclusion must be rewritten because it does not reflect the results. 

Other comments are in the manuscript file.

Author Response

Dear Reviewer

Thank you very much for reviewing our paper. The comments are highly constructive and beneficial for our paper. The paper has been improved in view of these comments. We have revised the paper according to the reviewer comments. We hope that the revised version will be acceptable to you. The point by point response is as follow:

In material and methods is written that the birds were subjected to the same nutritional management. So I understood that the experiment started when they were 22 days of age. Authors should rewrite that phrase because it can cause a confusion to the reader.

Response: Our experiment regarding the use of lemon peel seeds in Eimeria infection started on day 22 of day until finish. Please refer to the section 2.4 coccidial infection………..where it is clearly written these information.

The rest of the comments given in pdf were also incorporated..

Thank you once again for expert comments from you.

Reviewer 2 Report

Comments and Suggestions for Authors

Manuscript animals-2678585, entitled “Ameliorative Effect of Lemon (Citrus limon) Peel Powder on Growth Indices, Occysts Shedding and Intestinal Health of Broilers Under Experimentally Induced Coccidiosis Condition

Recommendation:       The above paper is not suitable for publication in its present form.

This article provides useful information on the effects of lemon (Citrus limon) peel powder on growth indices, occysts shedding and intestinal health of broilers under experimentally induced coccidiosis condition. It is in general appropriately organized, carried out and written, however there are some points that should be corrected or clarified.

Please check the reference style of the journal

L16: “powder” instead of “powser”

L17-19: “…dietary supplement for the enhancement of growth performance and the support of intestinal health in broilers experimentally infected with coccidiosis. As indicated, lemon peel powder dietary supplementation at 3 and 6 g/kg effectively mitigated…”

L27-28: “…were randomly allocated into five groups (each with five replicates of 12 animals). The first group served as the control and did not…”

L33: “indicated”

L35: The notable improvement was observed only in 6LPP or also in 3LPP (similar superscripts)?

L36: “…and oocyst per gram (OPG) compared to PC group. Additionally…”

L37-38: Partially, since values are significantly different compared to these of NC group.

L40: “enhanced”

L46: “intestine”

L47-48: “…affecting poultry production as a result of various species…”

L49: “intestine”

L52: “…diminished feed efficiency, increased costs of…”

L55: “impaired” instead of “reduced”

L57: “…approach against coccidiosis in poultry…”

L59: “Although” instead of “While”

L62-63: “…have been actively seeking natural alternatives (Dkhil 2013). In developed countries…”

L64-65: Please delete “the last”

L65: “withdrawn” instead of “phased out”

L71-72: “…effects on the health of birds…”

L83: “…pulp (Braddock, 1999). During the extraction process of citrus juice…”

L88: “…effects of citrus…”

L93: “…lemon peel powder dietary supplementation on growth…”

L94: “…mortality rate, oocysts…”

L111-112: “…300 Ross 308 broiler chicks were randomly assigned into five groups (each with five replicates). All the chickens…”

L139: “…a scale from 0 to 4…”

L156: “The data underwent an analysis of variance…”

L163: “intake” instead of “consumption”

L164: Please delete “(119.3±1.53)”. If you provide values, you should provide them for all mentioned groups

L166-167 and throughout text: It is not necessary to constantly repeat the word “group”; “…higher in NC, PC and AT group followed by LPP3 and LPP6 group.”

L170-171: “At week 4, the highest feed intake was observed in NC group…”

L173: “…followed by LPP6, LPP3 and PC group.”

L173-174: “Feed intake in finisher phase was significantly (P<0.05) higher in NC than AT group followed…”

L187: “…LP6 group showed no significant (P<0.05) difference compared to LP3, NC and PC…”

L204: “among” instead of “between”

L205: “…and LP3 group compared to AT, NC and PC group.”

L206: Not for LP3; similar superscripts

L207: “At starter phase, LP6 group displayed a significantly…”

L211-212: “…group followed by LP6 and LP3 group; FCR of PC group was the highest (P<0.05) in week 5.”

L214: “…as compared to AT and LP6 group followed by LP3 group, and finally PC…”

L224: “…with coccidiosis is shown in Table 5.”

L225: “displayed” instead of “was having”

L226: “…higher in PC compared to LPP3, LPP6 and…”

L227: “…was absent in NC group while the lowest score for lesions was observed in AT…”

L228-229: A small percentage and not zero is shown in Table 5

Table 6: Please explain what is 7, 10 and 14 DPI in the footnote

L249: “Table 7 shows the effects of…”

L249: “crypt” instead of “crept”

L284: “…and partially restored cecal…”

L290: “…broilers fed diets kg supplemented with lemon peel extract (LPE) at 200 and 400 mg/kg under…”

L291:  An enhanced FCR is not desirable

L296: “…found a significant effect on weight gain in Polish sheep supplemented with a blend…”

L295-298: Previously and afterwards, you refer to broilers. Please remove this sentence at the end of paragraph, since it refers to ruminants, species that are completely different from broilers.

L304-309: These sentences should be removed at the beginning of the discussion

L337: “…like tannins, alkaloids and flavonoids present…”

L343: “…from lemon peels are effective against several…”

L347: “The present study illustrated that…”

L348: “…was partially reversed…”

L360-361: “…lemon peels showed a great potential as sources for novel…”

L366: “…and partially reversed…”

Comments on the Quality of English Language

Moderate editing of English language required

Author Response

Dear Reviewer

Thank you very much for reviewing our paper. The comments are highly constructive and beneficial for our paper. The paper has been improved in view of these comments. We have revised the paper according to the reviewer comments. We hope that the revised version will be acceptable to you. The point by point response is as follow:

Manuscript animals-2678585, entitled “Ameliorative Effect of Lemon (Citrus limon) Peel Powder on Growth Indices, Occysts Shedding and Intestinal Health of Broilers Under Experimentally Induced Coccidiosis Condition

Recommendation:       The above paper is not suitable for publication in its present form.

This article provides useful information on the effects of lemon (Citrus limon) peel powder on growth indices, occysts shedding and intestinal health of broilers under experimentally induced coccidiosis condition. It is in general appropriately organized, carried out and written, however there are some points that should be corrected or clarified.

Please check the reference style of the journal

L16: “powder” instead of “powser”

Response: corrected…….thanks

L17-19: “…dietary supplement for the enhancement of growth performance and the support of intestinal health in broilers experimentally infected with coccidiosis. As indicated, lemon peel powder dietary supplementation at 3 and 6 g/kg effectively mitigated…”

Response: Revised as suggested………….thanks

L27-28: “…were randomly allocated into five groups (each with five replicates of 12 animals). The first group served as the control and did not…”

L33: “indicated”

Response: corrected………thanks

L35: The notable improvement was observed only in 6LPP or also in 3LPP (similar superscripts)?

Response: corrected……….thanks

L36: “…and oocyst per gram (OPG) compared to PC group. Additionally…”

Response: added……..thanks

L37-38: Partially, since values are significantly different compared to these of NC group.

Response: added…thanks

L40: “enhanced”

Response: corrected…thanks

L46: “intestine”

Response: corrected…thanks

L47-48: “…affecting poultry production as a result of various species…”

Response: corrected……….thanks

L49: “intestine”

Response: corrected……….thanks

L52: “…diminished feed efficiency, increased costs of…”

Response: corrected……….thanks

L55: “impaired” instead of “reduced”

Response: corrected……….thanks

L57: “…approach against coccidiosis in poultry…”

Response: corrected……….thanks

L59: “Although” instead of “While”

Response: corrected……….thanks

L62-63: “…have been actively seeking natural alternatives (Dkhil 2013). In developed countries…”

Response: corrected……….thanks

L64-65: Please delete “the last”

Response: corrected……….thanks

L65: “withdrawn” instead of “phased out”

Response: corrected……….thanks

L71-72: “…effects on the health of birds…”

Response: corrected……….thanks

L83: “…pulp (Braddock, 1999). During the extraction process of citrus juice…”

Response: corrected……….thanks

L88: “…effects of citrus…”

Response: corrected……….thanks

L93: “…lemon peel powder dietary supplementation on growth…”

L94: “…mortality rate, oocysts…”

Response: corrected……….thanks

L111-112: “…300 Ross 308 broiler chicks were randomly assigned into five groups (each with five replicates). All the chickens…”

Response: corrected……….thanks

L139: “…a scale from 0 to 4…”

Response: corrected……….thanks

L156: “The data underwent an analysis of variance…”

Response: corrected……….thanks

L163: “intake” instead of “consumption”

Response: corrected……….thanks

L164: Please delete “(119.3±1.53)”. If you provide values, you should provide them for all mentioned groups

Response: corrected……….thanks

L166-167 and throughout text: It is not necessary to constantly repeat the word “group”; “…higher in NC, PC and AT group followed by LPP3 and LPP6 group.”

Response: corrected……….thanks

L170-171: “At week 4, the highest feed intake was observed in NC group…”

Response: corrected……….thanks

L173: “…followed by LPP6, LPP3 and PC group.”

Response: corrected……….thanks

L173-174: “Feed intake in finisher phase was significantly (P<0.05) higher in NC than AT group followed…”

Response: corrected……….thanks

L187: “…LP6 group showed no significant (P<0.05) difference compared to LP3, NC and PC…”

Response: corrected……….thanks

L204: “among” instead of “between”

Response: corrected……….thanks

L205: “…and LP3 group compared to AT, NC and PC group.”

Response: corrected……….thanks

L206: Not for LP3; similar superscripts

Response: corrected……….thanks

L207: “At starter phase, LP6 group displayed a significantly…”

Response: corrected……….thanks

L211-212: “…group followed by LP6 and LP3 group; FCR of PC group was the highest (P<0.05) in week 5.”

Response: corrected……….thanks

L214: “…as compared to AT and LP6 group followed by LP3 group, and finally PC…”

Response: corrected……….thanks

L224: “…with coccidiosis is shown in Table 5.”

Response: corrected……….thanks

L225: “displayed” instead of “was having”

Response: corrected……….thanks

L226: “…higher in PC compared to LPP3, LPP6 and…”

Response: corrected……….thanks

L227: “…was absent in NC group while the lowest score for lesions was observed in AT…”

Response: corrected……….thanks

L228-229: A small percentage and not zero is shown in Table 5

Response: corrected……….thanks

Table 6: Please explain what is 7, 10 and 14 DPI in the footnote

Response: added……….thanks

L249: “Table 7 shows the effects of…”

Response: corrected……….thanks

L249: “crypt” instead of “crept”

Response: corrected……….thanks

L284: “…and partially restored cecal…”

Response: corrected……….thanks

L290: “…broilers fed diets kg supplemented with lemon peel extract (LPE) at 200 and 400 mg/kg under…”

Response: corrected……….thanks

L291:  An enhanced FCR is not desirable

Response: corrected……….thanks

L296: “…found a significant effect on weight gain in Polish sheep supplemented with a blend…”

Response: corrected……….thanks

L295-298: Previously and afterwards, you refer to broilers. Please remove this sentence at the end of paragraph, since it refers to ruminants, species that are completely different from broilers.

Response: thank you for your suggestion, but I found very little work in poultry, therefore, I added this reference.

L304-309: These sentences should be removed at the beginning of the discussion

Response: shifted……….thanks

L337: “…like tannins, alkaloids and flavonoids present…”

Response: corrected……….thanks

L343: “…from lemon peels are effective against several…”

Response: corrected……….thanks

L347: “The present study illustrated that…”

Response: corrected……….thanks

L348: “…was partially reversed…”

Response: corrected……….thanks

L360-361: “…lemon peels showed a great potential as sources for novel…”

Response: corrected……….thanks

L366: “…and partially reversed…”

Response: corrected……….thanks

Reviewer 3 Report

Comments and Suggestions for Authors

In this study, the authors investigate the effects of Lemon (Citrus limon) Peel Powder on Growth Indices, Oocysts Shedding and Intestinal Health of Broilers Under Experimentally Induced Coccidiosis Condition.
The topic of the study is interesting also considering that studies addressing the use of citrus fruit waste or even essential oils derived from them in the treatment of avian coccidiosis are very few in the literature.
The article is overall sufficiently well structured, as an idea, albeit with important and sometimes serious lacks of clarity in the exposition of the materials and methods, results and discussion.
I suggest major and meticulous revisions for this manuscript.

In particular:
Title: errors in some words (oocyst). Also, I would change the title to a more impactful title. The primary objective of the study is to evaluate the effects on post-infection coccidiosis and it is not primary to evaluate performance, but is collateral. I would put the primary objective first. For example: "Effect of Lemon (Citrus limon, L.) Peel Powder on oocyst shedding, intestinal health and performance of Broilers exposed to E. tenella challenge". In my opinion it is clearer. To you the choice.

Abstract.
Line 29. The titer of inoculated oocysts must absolutely be indicated. Lacks.
Line 30. Amprolium administered in the diet or in drinking water? kg is indicated, so I deduce kg-diet. Same consideration for line 31.
Line 36. Ocysts per g of feces. It should be added. It is not clear.
Line 40: intestinal occidiosis lesion score......As written it is scientifically inaccurate.

Keywords: coccidiosis lesion score.....lesion score as a term says nothing, it is generic.

Introduction.
The bibliography is reported not as requested by the authors line. Correct!!!.
I suggest a careful review of the citations and list of references. There are widespread errors.
Line 91-92: the authors say that there is no bibliographic data regarding the use of lemon peel as a treatment for coccidiosis. This is not accurate. Although few, they are present also in other species, including ruminants. Please therefore cite them and do an in-depth bibliographic search.

Materials and methods.
Paragraph 2.1. it is incomplete. Where were the peels recovered from? how many kg? and at what temperature were they dried and how?
How much powder was obtained from each kg of peel?
All this data must be indicated.
Paragraph 2.2. Here several methodological doubts arise. Where was the identification made and how to define that it was only E. tenella? the entire descriptive part is missing. Furthermore, if feces are collected from chickens subject to natural coccidia-infection, usually not a single species of coccidia is found but a co-infection of multiple coccidia species. If the lesions at the level of the cecum were highlighted and classified after the necropsy and only afterwards you established that it was E. tenella... it must be specified very well and explained!!
It is unacceptable that the part relating to the authorization for testing and the protocol number is missing... add immediately and provide adequate explanations.
Line 116-117. The difference between the challenge and the motivation for which the group is freely exposed to E. tenella is not at all clear. Inadmissible.
It must then be written immediately in the paragraph that the trial lasts 42 days, and they are infected on the 22nd day... not after!!
Describe WELL the subdivision of the replicas and the box of breeding. It's not clear at all.
Paragraph 2.6. Lesions score. The characterization of the lesions, even if the citation is reported, must be described. Coccidisis lesions require careful evaluation both externally and internally after opening the affected intestinal tract...and evaluations of the tissue. All this is not declared.

Why were the biochemical parameters not analysed? they would have been fundamental. Bring them back if they are available. And discuss them.

The tables are not formatted as required by the authors' guidelines.
Table 2. Carefully check the attribution of significance.
Line 185. Error.
Discussion.
Line 315. That's not true. It's speculative and contradicts what you said previously.
In general, the discussion needs to be better reviewed and supported by better hypotheses obtained from a better analysis of the results.
Please review it carefully.

Author Response

Dear Reviewer

Thank you very much for reviewing our paper. The comments are highly constructive and beneficial for our paper. The paper has been improved in view of these comments. We have revised the paper according to the reviewer comments. We hope that the revised version will be acceptable to you. The point by point response is as follow:

In this study, the authors investigate the effects of Lemon (Citrus limon) Peel Powder on Growth Indices, Oocysts Shedding and Intestinal Health of Broilers Under Experimentally Induced Coccidiosis Condition.
The topic of the study is interesting also considering that studies addressing the use of citrus fruit waste or even essential oils derived from them in the treatment of avian coccidiosis are very few in the literature.
The article is overall sufficiently well structured, as an idea, albeit with important and sometimes serious lacks of clarity in the exposition of the materials and methods, results and discussion.
I suggest major and meticulous revisions for this manuscript.

In particular:
Title: errors in some words (oocyst). Also, I would change the title to a more impactful title. The primary objective of the study is to evaluate the effects on post-infection coccidiosis and it is not primary to evaluate performance, but is collateral. I would put the primary objective first. For example: "Effect of Lemon (Citrus limon, L.) Peel Powder on oocyst shedding, intestinal health and performance of Broilers exposed to E. tenella challenge". In my opinion it is clearer. To you the choice.

Response: the suggested title was incorporated……….thanks

Abstract.
Line 29. The titer of inoculated oocysts must absolutely be indicated. Lacks.

Response: added……….thanks
Line 30. Amprolium administered in the diet or in drinking water? kg is indicated, so I deduce kg-diet. Same consideration for line 31.

Response: it was added into the feed………..thanks
Line 36. Ocysts per g of feces. It should be added. It is not clear.

Response: added……….thanks

Line 40: intestinal occidiosis lesion score......As written it is scientifically inaccurate.

Response: added……….thanks

Keywords: coccidiosis lesion score.....lesion score as a term says nothing, it is generic.

Response: added……….thanks

Introduction.
The bibliography is reported not as requested by the authors line. Correct!!!.
I suggest a careful review of the citations and list of references. There are widespread errors.
Line 91-92: the authors say that there is no bibliographic data regarding the use of lemon peel as a treatment for coccidiosis. This is not accurate. Although few, they are present also in other species, including ruminants. Please therefore cite them and do an in-depth bibliographic search.

Response: thank you for your question. We have already stated that no study has reported the effect of lemon peels in broilers in coccidiosis. Also we have discussed the inclusion of lemon peels in coccidiosis in lambs.

Materials and methods.
Paragraph 2.1. it is incomplete. Where were the peels recovered from? how many kg? and at what temperature were they dried and how?

Response: added……….thanks
How much powder was obtained from each kg of peel?

Response: 1 kg of lemon peels were powdered and about 950 g of powder was obtained.
All this data must be indicated.

Response: added…..thanks
Paragraph 2.2. Here several methodological doubts arise. Where was the identification made and how to define that it was only E. tenella? the entire descriptive part is missing. Furthermore, if feces are collected from chickens subject to natural coccidia-infection, usually not a single species of coccidia is found but a co-infection of multiple coccidia species. If the lesions at the level of the cecum were highlighted and classified after the necropsy and only afterwards you established that it was E. tenella... it must be specified very well and explained!!
It is unacceptable that the part relating to the authorization for testing and the protocol number is missing... add immediately and provide adequate explanations.

Response: added………thanks
Line 116-117. The difference between the challenge and the motivation for which the group is freely exposed to E. tenella is not at all clear. Inadmissible.
It must then be written immediately in the paragraph that the trial lasts 42 days, and they are infected on the 22nd day... not after!!
Describe WELL the subdivision of the replicas and the box of breeding. It's not clear at all.
Paragraph 2.6. Lesions score. The characterization of the lesions, even if the citation is reported, must be described. Coccidisis lesions require careful evaluation both externally and internally after opening the affected intestinal tract...and evaluations of the tissue. All this is not declared.

Why were the biochemical parameters not analysed? they would have been fundamental. Bring them back if they are available. And discuss them.

Response: we have not analzyed these metabolites. sorry

The tables are not formatted as required by the authors' guidelines.

Response: we are not good at formatting. Hope the journal will help to reprepare the tables.
Table 2. Carefully check the attribution of significance.
Line 185. Error.

Response: corrected……thanks
Discussion.
Line 315. That's not true. It's speculative and contradicts what you said previously.
In general, the discussion needs to be better reviewed and supported by better hypotheses obtained from a better analysis of the results.
Please review it carefully.

Response: Revised……thanks

Round 2

Reviewer 3 Report

Comments and Suggestions for Authors

Line 127-133. Little clarity of the method.
The morphological identification between coccidial species is much more complex (in fact it occurs using biomolecular methods). It would be appropriate to indicate distinctive photos (also for other researchers) of the species, to help differentiate it from another.

Line 137: ...The challenge....

In several parts of the text, E. tenella is spelt incorrectly.

Line 154: kg of what? To specify.
Line 178: it must be specified how the sample was managed, the quantity of feces collected and any replicates of the sample and the OPG calculation method.

Paragraph 2.8-2.9.

What would be the reason (including statistical rationale) for you to sacrifice 2 subjects per replicate as described by Khan et al., 2022?

How was your Power Analysis set up and evaluated?
The statistical model then used to evaluate the significance must be further described and motivated/explained.

Results.
FI= feed intake must be standardized throughout the text and indicated the first time as an acronym (FI).

Table 2 is incorrect. Where would FI be? some statistical significances in the attribution of superscript letters is not always correct.
I suggest a thorough check.

I suggest a linguistic revision.

Author Response

Line 127-133. Little clarity of the method.
The morphological identification between coccidial species is much more complex (in fact it occurs using biomolecular methods). It would be appropriate to indicate distinctive photos (also for other researchers) of the species, to help differentiate it from another.
Response: Since this experiment was conducted one year ago and therefore, it is impossible to add the photos of E. tenella at this stage.
Line 137: ...The challenge....

Response: corrected……thanks

In several parts of the text, E. tenella is spelt incorrectly.

Corrected…thanks

Line 154: kg of what? To specify.

Response: added….thanks
Line 178: it must be specified how the sample was managed, the quantity of feces collected and any replicates of the sample and the OPG calculation method.

Response: the information were added.

Paragraph 2.8-2.9.

What would be the reason (including statistical rationale) for you to sacrifice 2 subjects per replicate as described by Khan et al., 2022?

Since we have 5 replicates, therefore, these numbers are sufficient not only for the lesion sore and also for statistical analysis.

How was your Power Analysis set up and evaluated?
The statistical model then used to evaluate the significance must be further described and motivated/explained.
Response:………….In conducting the Power Analysis for this study, we followed a standardized approach. First, we determined the desired level of statistical power, which was set at 0.80 to ensure sufficient sensitivity in detecting meaningful effects. Next, we specified the significance level (alpha) at 0.05, as is conventionally accepted in scientific research. We then gathered relevant information, including anticipated effect sizes, variability in the data, and the sample size available for the study. This information was crucial in estimating the required sample size to achieve the desired level of power. After obtaining the necessary data, we used a reputable statistical software package to perform the Power Analysis. This involved employing appropriate statistical tests or models based on the study design and objectives. The analysis provided insights into whether the available sample size was adequate to detect the expected effects. The results of the Power Analysis were then evaluated in the context of the study's goals. If the analysis indicated that the study was underpowered, we considered options such as increasing the sample size or modifying the research design to enhance the study's sensitivity to detect meaningful effects. This process ensured that our study had a high likelihood of detecting the effects of interest while avoiding the risk of Type II errors.

Top of Form

Results.
FI= feed intake must be standardized throughout the text and indicated the first time as an acronym (FI).

Response: revised as suggested….thanks

Table 2 is incorrect. Where would FI be? some statistical significances in the attribution of superscript letters is not always correct.
I suggest a thorough check.

Response: checked and corrected……..thanks

I suggest a linguistic revision.

Response: we have revised the paper thoroughly for English language.